# Ecological Civilization Demonstration Zone, Air Pollution Reduction, and Political Promotion Tournament in China: Empirical Evidence from a Quasi-Natural Experiment

**DOI:** 10.3390/ijerph182211880

**Published:** 2021-11-12

**Authors:** Haijie Wang, Yong Geng, Jingxue Zhang, Xiqiang Xia, Yanchao Feng

**Affiliations:** 1Business School, Zhengzhou University, Zhengzhou 450001, China; whj@zzu.edu.cn (H.W.); m15803900638@163.com (J.Z.); xqxia@zzu.edu.cn (X.X.); 2School of Environmental Science and Engineering, Shanghai Jiao Tong University, Shanghai 200240, China; ygeng@sjtu.edu.cn

**Keywords:** ecological civilization demonstration zone, air pollution reduction, political promotion tournament, quasi-natural experiment, spatial difference-in-differences

## Abstract

Using the ecological civilization demonstration zone as a quasi-natural experiment, this study has explored the effect of it on air pollution in China by employing the difference-in-differences model and the spatial difference-in-differences model, and further tested the political promotion tournament in China by employing the binary logit model. The results show that the ecological civilization demonstration zone has basically and effectively reduced air pollution, except for carbon monoxide and ozone. In addition, the spatial spillover effects of the ecological civilization demonstration zone on air pollution are not only basically supported among the treated cities, but also extremely established in the untreated cities neighboring the treated cities. Furthermore, no clear evidence supports the establishment of the political promotion tournament in China, while local cadres tend to cope with the assessment of higher officials passively rather than actively. Overall, this study sheds light on the coordination of economic development and ecological civilization from the perspective of the career concerns of local cadres.

## 1. Introduction

As the undesirable by-product of industrialization and urbanization in China, environmental pollution not only hurts the human health of residents, but also threatens the sustainable development of biological systems [1]. In particular, air pollution has caused enough attention globally for its characteristic of spatial spillover effect [2]. Based on the data delivered by China’s Ecological Environment Status Bulletin in 2018, 64.2% of 338 prefecture-level and above cities exceeded air quality standards. Against this background, the joint governance of air pollution has become a consensus for policymakers, scholars, and the public [3], which forms the initial motivation of this study.

In the stage of shifting from high growth to high quality development, ecological civilization becomes a crucial issue in the new era [4]. As one representative nation of the centralized powers, the Chinese central government has a dominant role in environmental protection, where the top-down political promotion tournament system forced local cadres to focus on economic performance and infrastructure investment, while the governance of environmental pollution has been neglected in the past [5]. Fortunately, this situation began to change since the implementation of the 11th Five-Year Plan in 2006, with environmental performance incorporating to the promotion mechanism of local cadres [6]. In particular, “ecological civilization demonstration zones” (ECDZ), which are different from the prior economic zones, environmental protection zones, or free trade zones, become a breakthrough point [7,8].

Nevertheless, due to inconsistent environmental performance assessment standards across different regions, highly polluting enterprises tend to move from areas with strict control to areas with less control, which reduces the whole effect of environmental regulation and is detrimental to ecological protection [9]. Even so, air quality has been improved nationwide in recent years [10]. For instance, according to the data of China’s Ministry of Ecology and Environment, the PM_2.5_ non-compliance of prefecture-level and above cities in 2020 has decreased by 28.8% compared with 2015. However, whether and how ECDZ affects air pollution in China still remain as a black box, which leaves an opportunity for this study.

Due to the advantage of identifying the policy effect of quasi-natural experiments in econometric analysis, the difference-in-differences (DID) model has been widely used in policy evaluation [11]. Hence, in order to initially verify the effect of ECDZ on air pollution, this study attempts to set ECDZ as a multi-phrase quasi-natural experiment, and employ the DID model as the benchmark empirical method [11]. Furthermore, in order to identify the potential existence of spatial spillover effect, the spatial difference-in-differences (SDID) is also employed, especially in regional/spatial econometric studies [12]. Last but not least, in order to test the establishment of political promotion tournament in China, this study has established a comprehensive framework to investigate the impact of economic growth rate and air pollution reduction on political promotion by employing the binary logit model [13].

The marginal contributions of this study can be drawn from two aspects. Theoretically, the current estimates of the effects of air pollution are mainly concentrated in the economic and medical field, while few studies focused on the political field with the consideration of economic growth by establishing a comprehensive framework under the background of the political promotion tournament in China. Hence, this study takes advantage of the opportunity and has great theoretical significance for institutional economics and political economics, especially for the emerging countries with similar situations. In practice, the current discussion on ecological civilization mainly focuses on its measurement and driving forces, while no study has comprehensively explored the environmental impact of ECDZ in China, especially with the consideration of heterogeneous pollutants and spatial spillover effects. Therefore, this study has great practical significance for construction and optimization of ECDZ in China, and also has significant international meanings for other emerging powers.

We organize the rest of this study as follows. Section 2 gives a brief description of institutional background. Section 3 describes the data and method. Section 4 presents our results and proposes discussion. Section 5 concludes the main findings, delivers the policy implications, and points out the research prospects.

## 2. A Brief Description of Institutional Background

As a fundamental part of the reform, China’s central authority began to be decentralized to local governments in 1984, and this led to fierce fiscal competition among regional governments with the incentives of economic growth and political promotion [14]. On the one hand, although local governments share tax revenues with the central government, they are entirely responsible for local spending, while fiscal expenditure often exceeds revenue; thus off-budget revenue becomes the other channel to maintain high-speed economic growth [15]. On the other hand, to be recognized by the superior officials and gain political promotion under the top-down political system, local cadres tend to focus on the short-term goal of economic growth, which can be intuitively and quickly revealed by GDP and its growth [16].

However, since economic growth was placed in the primary position and the competition among local cadres was based on the economic performance, local governments tend to generally enforce environmental regulations with low standards to obtain mobility, which leads to extensive development at the expense of environmental pollution [17]. In particular, due to the characteristics of spatial spillover effect and regional heterogeneity for air pollution, the win of “blue sky defense war” still has a long way to go [18]. Against this background, ecological civilization, a new concept of green development, gradually gains the attention of policymakers and scholars [19].

As a philosophical topic and cultural theory, ecological civilization focused on balancing the relationship between humans and nature [20]. However, due to the limited space for science and technology discourse, ecological civilization has not been thoroughly conducted in practice until the administrative announcement of building it in the 17th Party Congress in 2007 [21]. Then, the construction of ecological civilization was incorporated into the overall layout of “Five-in-One” in the 18th Party Congress in 2012, and the concept of “beautiful China” highlighted the importance of accelerating the implementation of ecological civilization in the 19th Party Congress in 2017 [22]. Thus, in order to achieve the win–win of saving energy and reducing pollution, ecological civilization was gradually elevated to a more prominent position, which is also expected to be the endogenous driving force of green, sustainable, and high-quality development [23].

Against this background, to promote the construction of ecological civilization, the National Development and Reform Commission and other six ministries issued the “ecological civilization demonstration zone construction program (pilot)” in 2013, and then the first batch of ecological civilization demonstration zones (ECDZ) was officially designated in 2014, followed by the second batch of ECDZ in 2015 [24]. In particular, although the pilot areas of ECDZ formulated their own plans, respectively, the goal of cleaner production and green development is to some extent unified, which forms an ideal quasi-natural experiment to explore the effects of ECDZ on air pollution, and an opportunity to test the establishment of a political promotion tournament in China by investigating the effects of economic growth and air pollution reduction on political promotion.

To vividly illustrate two different types of promotion logic, we draw Figure 1 as follows. In the past, the traditional guiding ideology of a political promotion tournament was “only GDP”, that is, local cadres with better economic performance are more likely to be promoted while the opposite will face a lower promotion probability [16,17,18]. Since the announcement of ecological civilization, a new model of political promotion tournament was proposed and air pollution reduction (or environmental performance) had priority over economic growth, that is, only those who meet the environmental criteria firstly and achieve rapid economic growth simultaneously are more likely to be promoted [19,20,21,22,23]. Against this institutional reform, local cadres will have enough impetus in developing the economy and protecting the environment at the same time to achieve the win–win of an economy–environment balance.

## 3. Data and Methods

### 3.1. Data Sources

This study obtained the data from several sources. First, the air pollution data at the city level came from the Chinese Research Data Services (CNRDS) platform, which collects daily data on AQI, which was calculated by the concentrations of six air pollutants, namely, fine particulate matter with a diameter of 2.5 millionths of a meter or less (PM_2.5_), coarse particulate matter with a diameter of 10 millionths of a meter or less (PM_10_), sulfur dioxide (SO_2_), carbon monoxide (CO), nitrogen dioxide (NO_2_), and ozone (O_3_). Second, the biographical and promotional information of local cadres was mainly collected from two official websites, namely xinhuanet.com (accessed on 20 October 2020) and people.com (accessed on 21 October 2020). Third, the list of pilot cities where ECDZ were located was mainly sourced from the website of the National Development and Reform Commission in China. Fourth, the socioeconomic data were collected from the China City Statistic Yearbook and the China Urban Construction Statistic Yearbook. In particular, this study has deleted several cities, namely, the four municipalities directly under the central government (Beijing, Tianjin, Shanghai, and Chongqing), the prefecture level cities with severe data loss and administrative changes (Lhasa, Haidong, Turpan, Laiwu, and so on), and dozens of national autonomous prefecture cities, and eventually formed panel data including 280 cities from 2014 to 2019 as the studying samples.

### 3.2. Variables Measurement

The descriptive statistics of variables are reported in Table 1. In particular, the variables of this study have divided into two parts: Panel A and Panel B, which was employed to analyze the effects of ECDZ on air pollution and the effects of economic growth and air pollution reduction on political promotion, respectively.

In Panel A, the annual average value of AQI (AQI) and the concentrations of six air pollutants (i.e., PM_2.5_, PM_10_, SO_2_, CO, NO_2_, and O_3_) are adopted to act as the proxy of the dependent variables; DID, a dummy variable, denotes the core explanatory variable, which equals to 1 after the city owns the title of ECDZ, and otherwise 0; in addition, several control variables are introduced into the research framework to control for the characteristics of each city, including the natural logarithmic of per capita gross domestic product (lnPGDP) and its quadratic term ((lnPGDP)^2) to test the establishment of the Environmental Kuznets Curve, population density (PD) measured by the share of total population to municipal area, the share of secondary industry to GDP (SI), the share of third industry to GDP (TI), and financial pressure (FP) measured by the share of the difference between fiscal expenditure and fiscal revenue to GDP.

In panel B, political promotion (PP), a dummy variable, is adopted to act as the proxy of the dependent variables, which equals to 1 if the cadre is promoted to a higher rank, otherwise 0; economic growth rate with one year lag (EGR) and air pollution reduction (DAQI, DPM_2.5_, DPM_10_, DSO_2_, DCO, DNO_2_, and DO_3_) are employed to act as the proxy of the core explanatory variables; in addition, several control variables are employed to control for the characteristics of each cadre, including age (AGE), education (EDUCATION), gender (GENDER), and tenure (TENURE_1_ and TENURE_5_). Specifically, education (EDUCATION) is a dummy variable, which equals to 1 if the cadre has a doctorate degree, and 0 otherwise; gender (GENDER) is also a dummy variable, which equals to 1 if the cadre is a male, and 0 otherwise. Moreover, since the tenure of cadres is no more than 5 years, this study employs two dummy variables, namely TENURE_1_ and TENURE_5_, which equal to 1 if it is the first year or the last year of the tenure, and 0 otherwise.

### 3.3. Model Specification

Treating the establishment of ECDZ as a quasi-natural experiment, we first use the difference-in-differences (DID) model to evaluate the effects of it on air pollution in China [11], and construct the following basic model.
(1)$$Y_{it}=\beta_{0}+\beta_{1}\times DID_{it}+\beta_{2}\times X_{it}+\mu_{t}+\varepsilon_{it}$$
where Y_it_ represents the air pollution at city i in year t, β_0_ represents the constant term, DID_it_ represents the core explanatory variable, β_1_ represents the coefficient of DID_it_, X_it_ represents a vector of control variables, β_2_ represents the coefficients of X_it_, μ_t_ represents the time fixed effect, and ε_it_ represents the random error term.

The DID model based on a quasi-natural experiment can to some extent cause the endogeneity problem. However, the spatial spillover effects of air pollution cause the failing at obtaining consistent estimators under the traditional DID model. Fortunately, with the spatial lag terms incorporating into the traditional DID model, the spatial difference-in-differences (SDID) model can control such spatial spillover effects. In particular, considering the advantage of the SLX (i.e., a spatial lag of the independent variables) model in calculating local spillover effects rather than global spillover effects, this study has constructed the SDID model with the assumption of the SLX model [12], which is constructed as follows:(2)$$Y_{it}=\beta_{0}+\beta_{1}\times DID_{it}+\beta_{2}\times X_{it}+\theta_{1}\times W_{T,T}\times D_{it}+\theta_{2}\times W_{NT,T}\times D_{it}+\theta_{3}\times W\times X_{it}+\mu_{t}+\varepsilon_{it}$$
where W represents the first-order adjacency weight matrix, which is widely used in the empirical analysis of air pollution with a strong physical attribute [3], W_T,T_ × D_it_ represents the spatial spillover effects on the treated cities, θ_1_ represents the coefficient of W_T,T_ × D_it_, W_NT,T_ × D_it_ represents the spatial spillover effects on the untreated samples neighboring the treated cities, θ_2_ represents the coefficient of W_NT,T_ × D_it_, θ_3_ represents the spatial coefficient of X_it_, and the other parameters are consistent with Equation (1).

Since the dependent variable is discrete (i.e., 1 or 0) in panel B, the binary logit model is employed to test the establishment of political promotion tournament in China [13], dividing the research samples into two parts, namely the treated samples and the untreated samples, respectively.
(3)$$P_{it}=\alpha_{0}+\alpha_{1}\pi_{it}+\mu_{t}+\varepsilon_{it}$$
where P_it_ denotes political promotion at city i in year t, α_0_ denotes the constant term, π_it_ denotes the independent variables including air pollution reduction (DAQI, DPM_2.5_, DPM_10_, DSO_2_, DCO, DNO_2_, and DO_3_ incorporating into the equation one by one), economic growth rate (EGR) and the control variables, α_1_ is the coefficients of the independent variables, and the other parameters such as γ_i_, μ_t_, and ε_it_ are also consistent with Equation (1).

## 4. Empirical Results and Analysis

### 4.1. Diagnostic Tests of the DID Model

Before implementing the regression, two diagnostic tests including the parallel trend test [25] and the placebo test [3,26] have been conducted to guarantee the suitability and reliability of the DID model. Taking AQI, for example, the results for those two tests are shown in Figure 2 and Figure 3, respectively. It is clear that the parallel trend test and the placebo test are satisfied, thus the DID model is suitable for this study. Similarly, those two tests are also satisfied for other air pollutants.

### 4.2. Benchmark Regression Results

Based on Equation (1), this study has investigated the effects of ECDZ on the seven indicators of air pollution including AQI, PM_2.5_, PM_10_, SO_2_, CO, NO_2_, and O_3_, and reported the corresponding results in the columns (1)–(7) of Table 2, respectively.

As is shown in Table 2, the coefficients of DID in columns (2)–(4) and (6) are significantly negative, which indicates that ECDZ has reduced the annual average concentrations of PM_2.5_, PM_10_, SO_2_, and NO_2_. In addition, the coefficient of DID in column (1) is negative but insignificant, which indicates that the reduction effect of ECDZ on AQI is extremely weak in the statistics. However, the coefficients of DID in columns (5) and (7) are significantly positive, which indicates that ECDZ has increased the annual average concentrations of CO and O_3_. Thus, without the consideration of the spatial spillover effect, the effects of ECDZ on different indicators of air pollution have a clear differentiation; this is because compared with CO and O_3_, the other four indicators including PM_2.5_, PM_10_, SO_2_, and NO_2_ directly related to haze pollution, which are more sensitive to the public, and more effort is made to control them. Furthermore, long-term and short-term exposure to ambient ground level NO_2_ and O_3_ could play an important role in the phenotypes of cardio-respiratory diseases and inflammatory process in the lungs [27,28], thus the increased O_3_ and the reduced NO_2_ highlights the complex and difficulty of air pollution reduction in the process of constructing ecological civilization. Therefore, it is not hard to learn why local cadres tend to cope with the assessment of higher officials passively rather than actively.

### 4.3. Spatial Regression Results

Based on the Equation (2), this study has explored the spatial effects of ECDZ on the seven indicators of air pollution including AQI, PM_2.5_, PM_10_, SO_2_, CO, NO_2_, and O_3_, and reported the spatial regression results in the columns (1)–(7) of Table 3, respectively.

As is shown in Table 3, the coefficients of DID in columns (1)–(3) are significantly negative, the coefficients of it in columns (4) and (6) are negative but insignificant, while the coefficients of it in columns (5) and (7) are significantly positive, which are basically consistent with the benchmark results in Table 2 when ignoring the significance of the coefficients, implying the former effects of ECDZ on the seven indicators of air pollution are relatively robust in local cities. In addition, the spatial coefficients of W_T,T_D in columns (2)–(4), and (6) are significantly negative, the spatial coefficient of it in column (1) is negative but insignificant, while the coefficients of it in columns (5) and (7) are significantly positive, which indicates that ECDZ has indirectly reduced the annual average concentrations of PM_2.5_, PM_10_, SO_2_, and NO_2_, and increased the annual average concentrations of CO and O_3_ among the treated cities, while the reduction effect of it on AQI among the treated cities is relatively weak. Moreover, the spatial coefficients of W_NT,T_D in columns (1)–(4), and (6) are significantly negative, while the spatial coefficients of it in columns (5) and (7) are significantly positive, which indicates that ECDZ has indirectly reduced the annual average value of AQI and the annual average concentrations of PM_2.5_, PM_10_, SO_2_, and NO_2_, and increased the annual average concentrations of CO and O_3_ in the untreated cities neighboring the treated cities. Therefore, with the consideration of spatial spillover effects, the impacting trends of ECDZ on the seven indicators of air pollution are basically robust when compared with the benchmark results in Table 2, while the spatial spillover effects of ECDZ on the pilot and non-pilot cities have been identified precisely, which highlights the importance of superior governance once again.

### 4.4. Test on the Political Promotion Tournament in China

The effects of ECDZ on air pollution have been verified by adopting the DID and SDID models, while the establishment of the political promotion tournament in China has not been identified, which still needs an empirical test. Based on Equation (3), this study has investigated the effects of economic growth rate and air pollution reduction on political promotion in the treated samples and the untreated samples, and reported the results in Table 4.

As shown in Table 4, the coefficients of EGR are insignificantly positive in columns (1), (3), (5), (7), (9), (11), and (13), but significantly negative in columns (2), (4), (6), (8), (10), (12), and (14), indicating that economic growth rate has reduced the probability of political promotion in the untreated samples, but has a weak trend to promote the probability of political promotion in the treated samples. In addition, except for the significantly positive coefficient of DCO in column (9), all the other thirteen coefficients of air pollution reduction are insignificant in the statistics, that is, only the reduction in CO increases the probability of political promotion in the treated samples. One possible reason is that the governance of CO is easier to see the effect in a short time and in turn becomes the key work of local cadres, especially in the areas of ECDZ, which highlights that local cadres tend to cope with the assessment of higher officials passively rather than actively once again. Therefore, although the political promotion tournament in China is not supported in this study, the establishment of the new promotion mechanism based on attaching equal importance to economy and environment still has a long way to go.

## 5. Conclusions, Policy Implications, and Research Prospects

### 5.1. Conclusions

Based on the panel data of 280 Chinese cities from 2014 to 2019, this study has explored the effects of ECDZ on air pollution by adopting the DID and SDID models, and tested the establishment of the political promotion tournament in China by investigating the effects of economic growth rate and air pollution reduction on political promotion by adopting the binary logit model. This study draws the conclusions below.

First, except for CO and O_3_, ECDZ has to some extent reduced air pollution in China, no matter with or without the consideration of spatial spillover effects, which indicates that the effectiveness of ECDZ on reducing air pollution is basically established for the indicators of air pollution, namely, AQI, PM_2.5_, PM_10_, SO_2_, and NO_2_, while the controls of CO and O_3_ have been neglected, which deserves more attention in the future.

Second, except for the direct effects of ECDZ on air pollution, ECDZ has reduced the annual average values of AQI and the annual average concentrations of PM_2.5_, PM_10_, SO_2_, and NO_2_ among the treated cities and in the untreated cities neighboring the treated cities, but increased the annual average concentrations of CO and O_3_ among the treated cities and in the untreated cities neighboring the treated cities, which highlights the robustness of the impacting trend.

Third, although the political promotion tournament in China is not supported, the effectiveness of ECDZ has been to some extent proved. In particular, the governance of CO has indeed increased the possibility of political promotion in the treated cities. One possible reason is that the governance of CO is easier to see the effects of in the short-term and in turn becomes the key work of local cadres, which highlights that local cadres tend to cope with the assessment of higher officials passively rather than actively.

### 5.2. Policy Implications

In order to achieve the win-win of blue sky and economic development and promote the reform of political promotion mechanism simultaneously, several policy implications are provided as follows.

First, due to the differentiation of ECDZ on different indicators of air pollution, the traditional “one-size-fits-all” pattern should be changed, and the governance of them should be adapted to local conditions. In addition, to guarantee the effectiveness of environmental governance institutionally, the evaluation of environmental pollution and protection should be incorporated into the political promotion mechanism.

Second, due to the existence of spatial spillover effects and the collaborative governance problem caused by administrative segmentation, the joint governance of air pollution should be strengthened by adopting a top-level design. In addition, in order to improve the efficiency of cleaner production and lower the emissions of air pollutants simultaneously, a national exchange market of pollutants quota should be established soon.

Third, in order to stimulate the motivation of local cadres and reduce their short-sight behaviors, a lifelong accountability of ecological civilization should be established. In addition, to promote the institutionalization, normalization, and standardization of ECDZ, the institutional joint governance mechanism should cooperate with the joint of the experts in environmental, economic, engineering, and other fields.

### 5.3. Research Prospects

Since this study has offered a novel perspective on identifying the effects of ECDZ on air pollution and the establishment of political promotion tournament in China, several potential directions still need an in-depth exploration. For instance, due to the limited availability of data, the panel data of this study merely include 6 years, which may to some extent soften the persuasiveness. In addition, political connections may be incorporated into the research framework to test the establishment of rent-seeking (Kong et al., 2020). Therefore, with the increase in the possibility of available samples and indicators, this topic still deserves further investigation.

## Figures and Tables

**Figure 1 ijerph-18-11880-f001:**
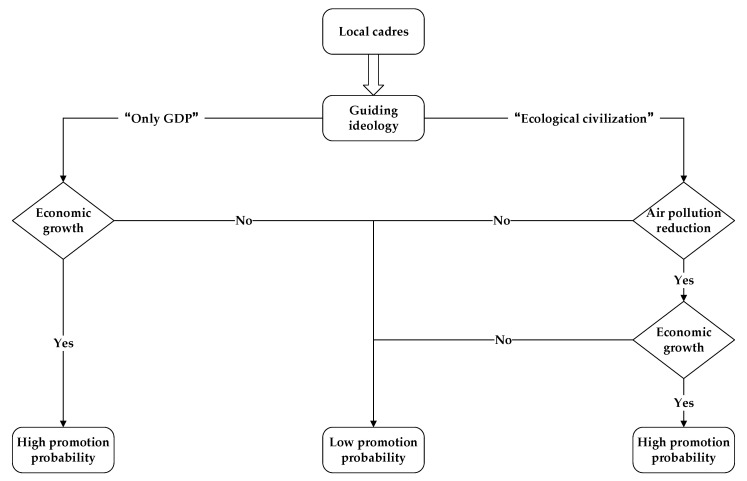
Two types of promotion logic in China.

**Figure 2 ijerph-18-11880-f002:**
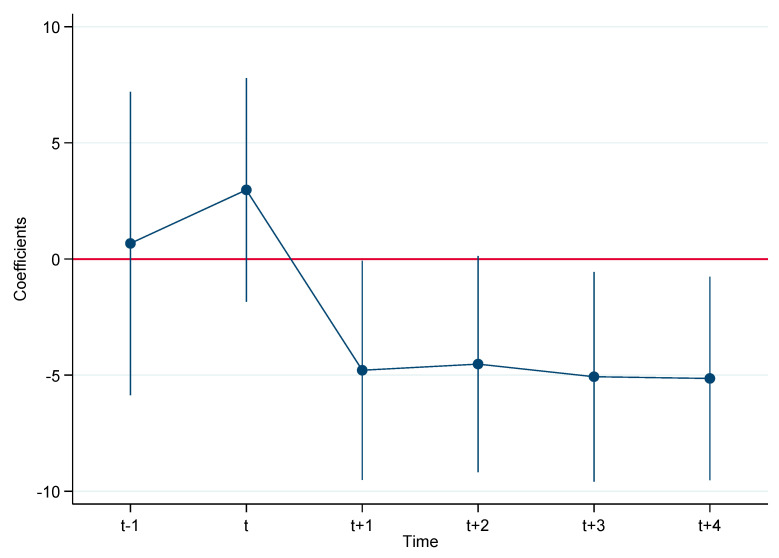
Parallel trend test (*AQI*).

**Figure 3 ijerph-18-11880-f003:**
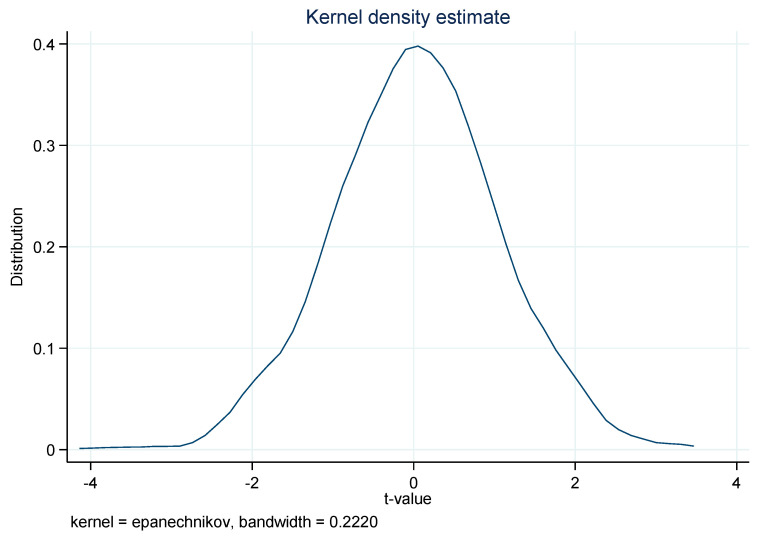
Placebo test (*AQI*).

**Table 1 ijerph-18-11880-t001:** Descriptive statistics.

Panel	Variables	Unit	Count	Mean	S.D	Min	Max
A	AQI	-	1680	80.208	22.978	18.000	230.000
	PM_2.5_	mg/m^3^	1680	46.770	18.857	9.000	150.000
	PM_10_	mg/m^3^	1680	81.776	32.507	18.000	371.000
	SO_2_	mg/m^3^	1680	21.841	17.011	2.000	185.000
	CO	mg/m^3^	1680	0.923	0.431	0.000	7.000
	NO_2_	mg/m^3^	1680	30.890	10.240	5.000	66.000
	O_3_	mg/m^3^	1680	87.841	15.842	23.000	160.000
	DID	-	1680	0.339	0.474	0.000	1.000
	lnPGDP	10^4^ Yuan	1680	4.607	2.537	0.283	28.991
	(lnPGDP)^2	10^4^ Yuan	1680	27.659	39.268	0.080	840.489
	PD	Persons/km^2^	1680	555.017	515.626	12.460	6814.815
	SI	%	1680	44.839	10.185	11.040	75.970
	TI	%	1680	45.032	10.400	19.770	79.710
	FP	%	1680	12.051	10.522	−13.660	92.650
B	PPT	-	1400	0.214	0.410	0.000	1.000
	GDP	%	1400	7.017	7.959	−38.663	136.163
	DAQI	-	1400	3.417	15.300	−64.580	146.580
	DPM_2.5_	mg/m^3^	1400	7.760	21.096	−70.580	276.000
	DPM_10_	mg/m^3^	1400	4.905	9.980	−30.250	128.170
	DSO_2_	mg/m^3^	1400	4.285	12.481	−52.580	92.250
	DCO	mg/m^3^	1400	−0.011	0.365	−1.840	5.770
	DNO_2_	mg/m^3^	1400	1.115	5.687	−24.500	39.250
	DO_3_	mg/m^3^	1400	−3.283	11.828	−68.000	69.250
	AGE	Year	1400	52.075	3.245	40.000	63.000
	EDUCATION	-	1400	0.779	0.415	0.000	1.000
	GENTER	-	1400	0.943	0.232	0.000	1.000
	TENURE_1_	-	1400	0.356	0.479	0.000	1.000
	TENURE_5_	-	1400	0.047	0.212	0.000	1.000

Notes: DAQI, DPM_2.5_, DPM_10_, DSO_2_, DCO, DNO_2_, and DO_3_ denotes the first-order difference of AQI, PM_2.5_, PM_10_, SO_2_, CO, NO_2_, and O_3_, respectively.

**Table 2 ijerph-18-11880-t002:** The baseline results based on the DID model.

Variables	AQI	PM_2.5_	PM_10_	SO_2_	CO	NO_2_	O_3_
	(1)	(2)	(3)	(4)	(5)	(6)	(7)
DID	−3.635	−7.131 ***	−8.814 **	−12.099 ***	0.372 ***	−1.772 *	10.601 ***
	(−1.423)	(−3.209)	(−2.383)	(−5.864)	(6.136)	(−1.836)	(4.958)
lnPGDP	−2.024 **	−3.854 ***	−5.555 ***	−2.235 ***	−0.001	−0.105	3.170 ***
	(−2.259)	(−4.942)	(−4.281)	(−3.087)	(−0.067)	(−0.309)	(4.226)
(lnPGDP)^2	0.044	0.115 ***	0.157 ***	0.093 ***	−0.000	−0.007	−0.128 ***
	(1.252)	(3.715)	(3.052)	(3.228)	(−0.182)	(−0.552)	(−4.320)
PD	−0.001	−0.002 **	−0.003 **	−0.002 ***	−0.000 ***	−0.001 **	0.002 **
	(−0.600)	(−2.038)	(−2.383)	(−2.816)	(−2.699)	(−2.561)	(2.233)
SI	−0.344 ***	−0.255 ***	−0.452 ***	−0.172 **	−0.009 ***	−0.158 ***	−0.208 **
	(−3.209)	(−2.733)	(−2.906)	(−1.979)	(−3.672)	(−3.894)	(−2.314)
TI	−0.918 ***	−0.947 ***	−1.676 ***	−0.961 ***	−0.012 ***	−0.281 ***	0.339 ***
	(−10.347)	(−12.268)	(−13.042)	(−13.402)	(−5.690)	(−8.378)	(4.559)
FP	−0.039	−0.095	−0.052	−0.078	0.005 ***	−0.026	0.057
	(−0.509)	(−1.427)	(−0.466)	(−1.266)	(2.923)	(−0.912)	(0.895)
Constant	147.130 ***	120.005 ***	204.285 ***	86.848 ***	1.739 ***	52.771 ***	65.538 ***
	(17.769)	(16.654)	(17.034)	(12.981)	(8.840)	(16.858)	(9.452)
Samples	1680	1680	1680	1680	1680	1680	1680
R-squared	0.159	0.265	0.280	0.320	0.074	0.106	0.151

Notes: t-statistics in parentheses, *** *p* < 0.01, ** *p* < 0.05, * *p* < 0.1.

**Table 3 ijerph-18-11880-t003:** The spatial results based on the SDID model.

Variables	AQI	PM_2.5_	PM_10_	SO_2_	CO	NO_2_	O_3_
	(1)	(2)	(3)	(4)	(5)	(6)	(7)
DID	−6.444 **	−6.606 **	−9.341 **	−3.895	0.344 ***	−1.445	7.181 ***
	(−1.966)	(−2.373)	(−2.020)	(−1.555)	(4.490)	(−1.158)	(2.648)
W_T,T_D	−7.696	−14.840 **	−25.962 **	−33.647 ***	0.782 ***	−5.144 *	19.311 ***
	(−0.962)	(−2.185)	(−2.302)	(−5.504)	(4.178)	(−1.689)	(2.918)
W_NT,T_D	−27.548 ***	−25.617 ***	−48.643 ***	−15.297 ***	1.005 ***	−6.588 ***	20.193 ***
	(−5.305)	(−5.810)	(−6.643)	(−3.855)	(8.273)	(−3.332)	(4.700)
Control	Yes	Yes	Yes	Yes	Yes	Yes	Yes
Samples	1680	1680	1680	1680	1680	1680	1680
R-squared	0.206	0.339	0.355	0.426	0.152	0.143	0.218

Notes: z-statistics in parentheses, *** *p* < 0.01, ** *p* < 0.05, * *p* < 0.1.

**Table 4 ijerph-18-11880-t004:** The promotion mechanism test.

Variables	Treated	Untreated	Treated	Untreated	Treated	Untreated	Treated	Untreated	Treated	Untreated	Treated	Untreated	Treated	Untreated
	(1)	(2)	(3)	(4)	(5)	(6)	(7)	(8)	(9)	(10)	(11)	(12)	(13)	(14)
EGR	0.015	−0.018 *	0.015	−0.018 *	0.014	−0.018 *	0.015	−0.018 *	0.014	−0.018 *	0.014	−0.018 *	0.014	−0.018 *
	(0.688)	(−1.670)	(0.679)	(−1.682)	(0.673)	(−1.675)	(0.693)	(−1.673)	(0.674)	(−1.709)	(0.674)	(−1.698)	(0.655)	(−1.680)
DAQI	−0.003	0.001												
	(−0.413)	(0.240)												
DPM_2.5_			−0.008	−0.012										
			(−0.612)	(−1.039)										
DPM_10_					−0.000	0.001								
					(−0.046)	(0.221)								
DSO_2_							−0.003	0.001						
							(−0.360)	(0.232)						
DCO									0.678 *	−0.118				
									(1.784)	(−0.496)				
DNO_2_											0.003	−0.010		
											(0.143)	(−0.709)		
DO_3_													−0.007	−0.001
													(−0.734)	(−0.138)
Control variables	Yes	Yes	Yes	Yes	Yes	Yes	Yes	Yes	Yes	Yes	Yes	Yes	Yes	Yes
Samples	505	895	505	895	505	895	505	895	505	895	505	895	505	895
R-squared	0.065	0.023	0.066	0.024	0.064	0.023	0.065	0.023	0.070	0.023	0.064	0.023	0.065	0.023

Notes: t-statistics in parentheses, * *p* < 0.1.

## Data Availability

The data used to support the findings of this study are available from the corresponding author upon request.

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
