# Peer review of "Ecological Civilization Demonstration Zone, Air Pollution Reduction, and Political Promotion Tournament in China: Empirical Evidence from a Quasi-Natural Experiment"

_ijerph, 2021, doi:10.3390/ijerph182211880_

Round 1

Reviewer 1 Report

This paper used the difference-in-difference model, spatial difference-in-difference model, and binary logit model to study the ecological civilization demonstration zone, air pollution reduction, and political promotion tournament in China. A panel data including 280 cities in China from 2014 to 2019 are used in the study. The difference-in-difference model results show that ecological civilization demonstration zone has significantly reduced PM2.5, PM10, SO2, and NO2, and significantly increased CO and O3. The spatial difference-in-difference model results show that ecological civilization demonstration zone has significantly reduced PM2.5 and PM10, reduced (but not statistically significant) SO2, and NO2, and significantly increased O3 and increased (but not statistically significant) CO. The binary logit model results show that there is no clear evidence supports the establishment of the political promotion tournament in China.

This is an interesting paper. The research findings have important contributions to the literature and the implications.  

  1. Please avoid acronyms in the abstract and spell out the DID and SDID in the abstract.

  1. On page 5, line 201

“Before implementing the regression, the parallel trend test and the placebo test have been conducted and supported by referring to the research of Feng et al. [3], which guarantees the reliability of this study.”

Suggest to revise this sentence to:

The difference-in-differences model (Angrist and Pischke, 2008) is used as the benchmark empirical method identify the potential existence of spatial spillover effect by the spatial difference-in-differences (SDID) method. The DID mode has been widely used in the estimation of a causal effect of a treatment by making use of longitudinal data from a treatment and a control group (Angrist and Pischke, 2008). The DID model is based on the parallel trend assumption. Therefore, a parallel trend test (Angrist and Pischke, 2008) and a placebo test (Chen, Hong, and Liu, 2017) are used to ensure the validity of the DID method in our research setting. The test results are shown in Figure X and Table X. Both of these tests show that …….. 

Figure X

Parallel trend analysis

Table X

Parallel Trend Placebo Test

Angrist, J. D. and Pischke, J. S. (2008). Parallel worlds: fixed effects, differences-in-differences, and panel data. In Mostly harmless econometrics (pp. 221-248). Princeton University Press.

Chen, P. Y., Hong, Y., Liu, Y. (2018). The value of multidimensional rating systems: Evidence from a natural experiment and randomized experiments. Management Science, 64(10), 4629-4647.

  1. Page 6, line 214

“the effects of ECDZ on different indicators of air pollution have a clear differentiation, this is because compared with CO and O3, the other four indicators including PM2.5, PM10, SO2, and NO2 directly related to haze pollution, which are more sensitive to the public, and more effort is made to control them.”

Suggest to discuss more about the ecological civilization demonstration zone has significantly increased CO and O3. The ground level O3 is the main ingredient in smog. Long-term and short-term exposure to ambient ground level O3 and NO2 can play an important role in the phenotypes of cardio-respiratory diseases and inflammatory process in the lungs (Jones, 2020; Zoran et al, 2020). Will these results affect the local cadre performance evaluation? If the answer is “no”. Then these will reinforce the argument of this paper.

The difference-in-difference model results show that ecological civilization demonstration zone has significantly reduced NO2, and significantly increased O3. But the ground level O3 could be created by chemical reactions between NO2 and volatile organic compounds. Why NO2 reduced and O3 increased? Is it because that NO2 had been transformed into O3 by chemical reactions?    

Jones, J. W. (2020). Evaluating changes in ambient ozone and respiratory-related healthcare utilization in the Washington, DC metropolitan area. Environmental Research, 186, 109603.

Zoran, M. A., Savastru, R. S., Savastru, D. M., and Tautan, M. N. (2020). Assessing the relationship between ground levels of ozone (O3) and nitrogen dioxide (NO2) with coronavirus (COVID-19) in Milan, Italy. Science of The Total Environment, 740, 140005.

  1. Equation 3 is used to test the factors that can affect the political promotion. Would it be better to use the previous year’s data for the independent variables? Because by the time of promotion, the same year independent data are not available yet.

Author Response

Responses to Reviewer # 1

Comments and Suggestions for Authors

This paper used the difference-in-difference model, spatial difference-in-difference model, and binary logit model to study the ecological civilization demonstration zone, air pollution reduction, and political promotion tournament in China. A panel data including 280 cities in China from 2014 to 2019 are used in the study. The difference-in-difference model results show that ecological civilization demonstration zone has significantly reduced PM2.5, PM10, SO2, and NO2, and significantly increased CO and O3. The spatial difference-in-difference model results show that ecological civilization demonstration zone has significantly reduced PM2.5 and PM10, reduced (but not statistically significant) SO2, and NO2, and significantly increased O3 and increased (but not statistically significant) CO. The binary logit model results show that there is no clear evidence supports the establishment of the political promotion tournament in China.

This is an interesting paper. The research findings have important contributions to the literature and the implications.  

Please avoid acronyms in the abstract and spell out the DID and SDID in the abstract.

Reply: Many thanks for your constructive suggestion, we have replace “the DID and SDID models” with “the difference-in-differences model and the spatial difference-in-differences model.

On page 5, line 201

“Before implementing the regression, the parallel trend test and the placebo test have been conducted and supported by referring to the research of Feng et al. [3], which guarantees the reliability of this study.”

Suggest to revise this sentence to:

The difference-in-differences model (Angrist and Pischke, 2008) is used as the benchmark empirical method identify the potential existence of spatial spillover effect by the spatial difference-in-differences (SDID) method. The DID mode has been widely used in the estimation of a causal effect of a treatment by making use of longitudinal data from a treatment and a control group (Angrist and Pischke, 2008). The DID model is based on the parallel trend assumption. Therefore, a parallel trend test (Angrist and Pischke, 2008) and a placebo test (Chen, Hong, and Liu, 2017) are used to ensure the validity of the DID method in our research setting. The test results are shown in Figure X and Table X. Both of these tests show that …….. 

Figure X

Parallel trend analysis

Table X

Parallel Trend Placebo Test

Angrist, J. D. and Pischke, J. S. (2008). Parallel worlds: fixed effects, differences-in-differences, and panel data. In Mostly harmless econometrics (pp. 221-248). Princeton University Press.

Chen, P. Y., Hong, Y., Liu, Y. (2018). The value of multidimensional rating systems: Evidence from a natural experiment and randomized experiments. Management Science, 64(10), 4629-4647.

Reply: Many thanks for your constructive suggestion, we have rewritten this section and added the referred literature after a careful reading.

4.1. Diagnostic tests of the DID model

Before implementing the regression, two diagnostic tests including the parallel trend test [25] and the placebo test [3, 26] have been conducted to guarantee the suitability and reliability of the DID model. Taken AQI for example, the results for those two tests are shown in Figure 2 and Figure 3, respectively. It’s clear that the parallel trend test and the placebo test are satisfied, thus the DID model is suitable for this study. Similarly, those two tests are also satisfied for other air pollutants.

Figure 2. Parallel trend test (AQI).

Figure 3. Placebo test (AQI).

Page 6, line 214

“the effects of ECDZ on different indicators of air pollution have a clear differentiation, this is because compared with CO and O3, the other four indicators including PM2.5, PM10, SO2, and NO2 directly related to haze pollution, which are more sensitive to the public, and more effort is made to control them.”

Suggest to discuss more about the ecological civilization demonstration zone has significantly increased CO and O3. The ground level O3 is the main ingredient in smog. Long-term and short-term exposure to ambient ground level O3 and NO2 can play an important role in the phenotypes of cardio-respiratory diseases and inflammatory process in the lungs (Jones, 2020; Zoran et al, 2020). Will these results affect the local cadre performance evaluation? If the answer is “no”. Then these will reinforce the argument of this paper.

The difference-in-difference model results show that ecological civilization demonstration zone has significantly reduced NO2, and significantly increased O3. But the ground level O3 could be created by chemical reactions between NO2 and volatile organic compounds. Why NO2 reduced and O3 increased? Is it because that NO2 had been transformed into O3 by chemical reactions?    

Jones, J. W. (2020). Evaluating changes in ambient ozone and respiratory-related healthcare utilization in the Washington, DC metropolitan area. Environmental Research, 186, 109603.

Zoran, M. A., Savastru, R. S., Savastru, D. M., and Tautan, M. N. (2020). Assessing the relationship between ground levels of ozone (O3) and nitrogen dioxide (NO2) with coronavirus (COVID-19) in Milan, Italy. Science of The Total Environment, 740, 140005.

Reply: Many thanks for your constructive suggestion, we have rewritten this section and added the referred literature after a careful reading.

Equation 3 is used to test the factors that can affect the political promotion. Would it be better to use the previous year’s data for the independent variables? Because by the time of promotion, the same year independent data are not available yet.

Reply: Many thanks for your constructive suggestion, we have replaced the current economic growth rate with the one year lag economic growth rate, and found new evidence to support the effectiveness of ECDZ.

Best regards,

Haijie Wang1, Yong Geng2, Jingxue Zhang1, Xiqiang Xia1, Yanchao Feng1,*

1 Business School, Zhengzhou University, Zhengzhou 450001, PR China

2 School of Environmental Science and Engineering, Shanghai Jiao Tong University, Shanghai 200240, PR China

Reviewer 2 Report

The manuscript presents an interesting topic looking at the impact of the ecological civilization demonstration zone on air pollution in China. I think this is a good fit for the International Journal of Environmental Research and Public Health. However, prior to being ready to be published in the journal, the authors need to consider addressing the following comments carefully:

  1. the authors need to add relevant literature on the empirical evaluation of environmental "zone" on air pollution - though the existing studies may not provide the evaluation on the same policy, there are many similar studies on similar "environmental zones" across different countries.
  2. the authors need to include relevant studies on the empirical models adopted to state the appropriateness of employing the models in the empirical analysis. At the current form, they did not provide any relevant literature on the use of SDID in the context of environmental studies/environmental economics studies.'
  3. as for the method part, followed by point 2, (1) the authors need to justify the use of SDID to the empirical analysis, for instance, why include spatial effects, why to use three spatial weights matrices, and what does the coefficient represents, respectively? (2) the authors need to specify why including the spatial effect can address the endogenous issue, as what they included are only for the independent variables rather than the lagged dependent variable (which is usually the endogenous part of a spatial model).
  4. once they have addressed the issues in the methods part, they can adjust their results and discussion, accordingly.

Author Response

Responses to Reviewer # 2

Comments and Suggestions for Authors

The manuscript presents an interesting topic looking at the impact of the ecological civilization demonstration zone on air pollution in China. I think this is a good fit for the International Journal of Environmental Research and Public Health. However, prior to being ready to be published in the journal, the authors need to consider addressing the following comments carefully:

Reply: Many thanks for your positive comments, we have addressed the following comments carefully.

the authors need to add relevant literature on the empirical evaluation of environmental "zone" on air pollution - though the existing studies may not provide the evaluation on the same policy, there are many similar studies on similar "environmental zones" across different countries.

Reply: Many thanks for your constructive comments. Indeed, there are many other “zones” including economic zones, environmental protection zones, and free trade zones, which have been explored by other scholars. We have added relevant literature and highlighted the evolution track of ECDZ in the introduction.

the authors need to include relevant studies on the empirical models adopted to state the appropriateness of employing the models in the empirical analysis. At the current form, they did not provide any relevant literature on the use of SDID in the context of environmental studies/environmental economics studies.'

Reply: Many thanks for your constructive comments, we have rewritten this section and added the use of the SDID model.

as for the method part, followed by point 2, (1) the authors need to justify the use of SDID to the empirical analysis, for instance, why include spatial effects, why to use three spatial weights matrices, and what does the coefficient represents, respectively? (2) the authors need to specify why including the spatial effect can address the endogenous issue, as what they included are only for the independent variables rather than the lagged dependent variable (which is usually the endogenous part of a spatial model).

once they have addressed the issues in the methods part, they can adjust their results and discussion, accordingly.

Reply: Many thanks for your constructive comments, we have addressed the issues you in the methods part, and adjusted their results and discussion, accordingly.

Best regards,

Haijie Wang1, Yong Geng2, Jingxue Zhang1, Xiqiang Xia1, Yanchao Feng1,*

1 Business School, Zhengzhou University, Zhengzhou 450001, PR China

2 School of Environmental Science and Engineering, Shanghai Jiao Tong University, Shanghai 200240, PR China

Reviewer 3 Report

Abstract:

The authors explain the main variables and outcome of the study, it is important to explain the objective and novelty of the study in the abstract.

Introduction:

In this section, the authors need to clarify the choice of variables and possible contributions to the current literature.

In practice, the current discussion on ecological civilization mainly focuses on its measurement and driving forces, while no study has explored the environmental impact of ECDZ, not alone to say the heterogeneity of pollutants and space.

The authors need to double-check this claim, if it’s true, why is that?

No theoretical framework, please add the theoretical framework to understand the mechanism through variables are interact and linked them with economic theories.

Data

Data is limited to 2014 to 2019 so based on this data how authors give policy implications are very limited.

Model Specification

The authors need to explain why they employed DID, SDID methods.

Results

The authors just explained the results column-wise, they require to compare these results with existing and explain their viewpoint, why they are positive, negative, or insignificant.

Conclusion  

The author needs to revise this section. The conclusion section should highlight more clearly how the results compare with the recent studies. Conclusions should consist of theoretical implications, managerial implications, limitations, and future research perspectives.

Author Response

Responses to Reviewer # 3

Comments and Suggestions for Authors

Abstract:

The authors explain the main variables and outcome of the study, it is important to explain the objective and novelty of the study in the abstract.

Reply: Many thanks for your constructive comments, we have rewritten the abstract and highlighted the novelty in this section, that is, this study sheds light on the coordination of economic development and ecological civilization from the perspective of the career concerns of local cadres.

Introduction:

In this section, the authors need to clarify the choice of variables and possible contributions to the current literature.

Reply: Many thanks for your constructive comments, we have rewritten the introduction, clarified the choice of variables, and highlighted the marginal contributions from the dual perspectives of theory and practice.  

In practice, the current discussion on ecological civilization mainly focuses on its measurement and driving forces, while no study has explored the environmental impact of ECDZ, not alone to say the heterogeneity of pollutants and space.

The authors need to double-check this claim, if it’s true, why is that?

Reply: Indeed, it is not proper to express this content, which may lead to misunderstandings and even mistakes, thus we have rewritten this place as follows:

In practice, the current discussion on ecological civilization mainly focuses on its measurement and driving forces, while no study has comprehensively explored the environmental impact of ECDZ in China, especially with the consideration of heterogeneous pollutants and spatial spillover effects.

No theoretical framework, please add the theoretical framework to understand the mechanism through variables are interact and linked them with economic theories.

Reply: Many thanks for your constructive comments, we have added the theoretical framework to highlight the logic of this study and linked them with economic theories as follows:

To illustrate two different types of promotion logic vividly, we draw Figure 1 as follows. In the past, the traditional guiding ideology of political promotion tournament was “only GDP”, that is, local cadres with better economic performance are more likely to be promoted while the opposite will face a lower promotion probability [16-18]. Since the announcement of ecological civilization, a new model of political promotion tournament was proposed and air pollution reduction (or environmental performance) had priority over economic growth, that is, only those who meet the environmental criteria firstly and achieve rapid economic growth simultaneously are more likely to be promoted [19-23]. Against this institutional reform, local cadres will have enough impetus in developing the economy and protecting the environment at the same time to achieve the win-win of an economy-environment balance.

Figure 1. Two types of promotion logic in China.

Data

Data is limited to 2014 to 2019 so based on this data how authors give policy implications are very limited.

Reply: Indeed, in order to overcome the limitation of this study, a follow-up study is necessary if more data is available in the future, and we point out this in the section of research prospects.

Model Specification

The authors need to explain why they employed DID, SDID methods.

 Reply: Many thanks for your constructive comments, we have addressed the necessity of the DID and SDID models in the introduction.

Results

The authors just explained the results column-wise, they require to compare these results with existing and explain their viewpoint, why they are positive, negative, or insignificant.

Reply: Many thanks for your constructive comments, we have addressed those issues and added the comparison with existing literature.

Conclusion  

The author needs to revise this section. The conclusion section should highlight more clearly how the results compare with the recent studies. Conclusions should consist of theoretical implications, managerial implications, limitations, and future research perspectives.

Reply: Many thanks for your constructive comments, we have rewritten the final section to highlight the main findings compared with the recent studies, and pointed out the policy implications, limitations, and research perspectives.

Best regards,

Haijie Wang1, Yong Geng2, Jingxue Zhang1, Xiqiang Xia1, Yanchao Feng1,*

1 Business School, Zhengzhou University, Zhengzhou 450001, PR China

2 School of Environmental Science and Engineering, Shanghai Jiao Tong University, Shanghai 200240, PR China

Reviewer 4 Report

Dear Authors,
The study aims at investigating the "Ecological civilization demonstration zone, air pollution reduction, and political promotion tournament in China: Empirical evidence from a quasi-natural experiment". 
The paper is not satisfactory written, needs a careful editing, fonts, and style. 
However, it is recommended:
- Reformulate the abstract by telling prospective readers what you did and what the important findings of your research were. I suggest not to use acronyms in the abstract.
- Introduction can be improved in order to show better aim. Is very short.
- Please carefully consider and revise the logic of some parts. 
- Carefully check the full text. 
- Further, the study aim and background are not well presented, repetitions occurring in the paper should be avoided. 
- Literature coverage in terms of papers is not balanced, references are classic. In terms of literature review, the paper does, unfortunately, take no unique or critical point of view. The paper should better provide a critical analysis of the available and appropriate literature.
- I suggest considering a general / integrative theoretical approach in order to present your research model, and then write your paper from the angle of the chosen one.
- Please do more to highlight how the work advances or increments the field from the present state of knowledge and provide a clear justification for your work.
- The methodological approach used is not clear. Seems like a statistical exercise not an econometric approach.
- Results must be rewritten.
- Conclusion section needs improvement. Please provide more quantitative key contributions of the study with proper discussions, highlight critically the limitations of this study and the future work. 
- English proofreading is needed. Some description is not professional for a scientific article.
Accordingly, it is opinion of this reviewer to accept with major revisions the proposed manuscript for a publication on this journal.

Author Response

Responses to Reviewer # 4

Comments and Suggestions for Authors

Dear Authors,
The study aims at investigating the "Ecological civilization demonstration zone, air pollution reduction, and political promotion tournament in China: Empirical evidence from a quasi-natural experiment". 
The paper is not satisfactory written, needs a careful editing, fonts, and style. 

 Reply: Many thanks for your constructive comments, we have conducted a careful editing, fonts, and style under the guide of your suggestions.

However, it is recommended:
- Reformulate the abstract by telling prospective readers what you did and what the important findings of your research were. I suggest not to use acronyms in the abstract.

 Reply: Many thanks for your constructive suggestion, we have replace the acronyms with their full names.

- Introduction can be improved in order to show better aim. Is very short.

 Reply: Many thanks for your constructive suggestion, we have rewritten the introduction to highlight the logic and aims of this study.

- Please carefully consider and revise the logic of some parts. 

- Carefully check the full text. 

 Reply:Many thanks for your constructive suggestion, we have checked the logic of the full text and revised several places including the introduction, literature review, methods, results, and conclusions.

- Further, the study aim and background are not well presented, repetitions occurring in the paper should be avoided. 

 Reply: Many thanks for your constructive suggestion, we have checked the corresponding places and corrected the repetitions.

- Literature coverage in terms of papers is not balanced, references are classic. In terms of literature review, the paper does, unfortunately, take no unique or critical point of view. The paper should better provide a critical analysis of the available and appropriate literature.

 Reply: Many thanks for your constructive suggestion, we have rewritten the first and second sections and reorganized the text with a critical analysis of the available and appropriate literature.

- I suggest considering a general / integrative theoretical approach in order to present your research model, and then write your paper from the angle of the chosen one.

 Reply: Many thanks for your constructive suggestion, we have added the theoretical framework to highlight the logic of this study and linked them with economic theories as follows:

To illustrate two different types of promotion logic vividly, we draw Figure 1 as follows. In the past, the traditional guiding ideology of political promotion tournament was “only GDP”, that is, local cadres with better economic performance are more likely to be promoted while the opposite will face a lower promotion probability [16-18]. Since the announcement of ecological civilization, a new model of political promotion tournament was proposed and air pollution reduction (or environmental performance) had priority over economic growth, that is, only those who meet the environmental criteria firstly and achieve rapid economic growth simultaneously are more likely to be promoted [19-23]. Against this institutional reform, local cadres will have enough impetus in developing the economy and protecting the environment at the same time to achieve the win-win of an economy-environment balance.

Figure 1. Two types of promotion logic in China.

- Please do more to highlight how the work advances or increments the field from the present state of knowledge and provide a clear justification for your work.

 Reply:Many thanks for your constructive suggestion, we have pointed out the marginal contributions of this study from the dual perspectives of theory and practice in the introduction section.

- The methodological approach used is not clear. Seems like a statistical exercise not an econometric approach.

 Reply: Many thanks for your constructive suggestion, we have rewritten the methodological approach in the first and third sections.

- Results must be rewritten.

 Reply: Many thanks for your constructive suggestion, we have rewritten the results and given new explanation.

- Conclusion section needs improvement. Please provide more quantitative key contributions of the study with proper discussions, highlight critically the limitations of this study and the future work. 

 Reply: Many thanks for your constructive comments, we have rewritten the final section to highlight the main findings compared with the recent studies, and pointed out the policy implications, limitations, and research perspectives.

- English proofreading is needed. Some description is not professional for a scientific article.

Reply: Many thanks for your constructive suggestion, we have rewritten this paper and invited two English native experts to improve the academic level of this study.

Accordingly, it is opinion of this reviewer to accept with major revisions the proposed manuscript for a publication on this journal.

 Reply: Many thanks for your positive comments. We hope that the revised revision has addressed all the issues. We are looking forward to your positive response. If you have any queries, please don’t hesitate to deliver your new comments. Best wishes for you and God bless you.

Best regards,

Haijie Wang1, Yong Geng2, Jingxue Zhang1, Xiqiang Xia1, Yanchao Feng1,*

1 Business School, Zhengzhou University, Zhengzhou 450001, PR China

2 School of Environmental Science and Engineering, Shanghai Jiao Tong University, Shanghai 200240, PR China

Reviewer 5 Report

The topic of the paper is very interesting and the authors have argued it in a proper manner. The major minus of this study is related to the conclusions: they should be developed, especially by including more the policy recommendations.

Author Response

Responses to Reviewer # 5

Comments and Suggestions for Authors

The topic of the paper is very interesting and the authors have argued it in a proper manner. The major minus of this study is related to the conclusions: they should be developed, especially by including more the policy recommendations.

 Reply: Many thanks for your positive comments. Even so, we have rewritten this paper and invited two English native experts to improve the academic level of this study, and we hope that the revised revision has addressed all the issues. We are looking forward to your positive response. If you have any queries, please don’t hesitate to deliver your new comments. Best wishes for you and God bless you.

Best regards,

Haijie Wang1, Yong Geng2, Jingxue Zhang1, Xiqiang Xia1, Yanchao Feng1,*

1 Business School, Zhengzhou University, Zhengzhou 450001, PR China

2 School of Environmental Science and Engineering, Shanghai Jiao Tong University, Shanghai 200240, PR China

Round 2

Reviewer 2 Report

The authors have addressed my comments carefully.

Reviewer 4 Report

Dear Authors,

I very much appreciate the efforts of the authors to meet my comments and suggestions and to implement the suggestions, observations and recommendations I made.

I am happy to inform you that I have accepted your revision of the manuscript and will recommend it for publication without further changes. Congratulations. I look forward to reading it online.

Thank you for the opportunity to let me contribute a small part to your publication.